# *Desulfovibrio* in the Gut: The Enemy within?

**DOI:** 10.3390/microorganisms11071772

**Published:** 2023-07-07

**Authors:** Sudha B. Singh, Amanda Carroll-Portillo, Henry C. Lin

**Affiliations:** 1Biomedical Research Institute of New Mexico, Albuquerque, NM 87108, USA; 2Division of Gastroenterology and Hepatology, University of New Mexico, Albuquerque, NM 87131, USA; acarrollportillo@salud.unm.edu; 3Medicine Service, New Mexico VA Health Care System, Albuquerque, NM 87108, USA

**Keywords:** *Desulfovibrio*, sulfate reducing bacteria (SRB), Parkinson’s disease, inflammatory bowel disease, hydrogen sulfide (H_2_S), lipopolysaccharide (LPS)

## Abstract

*Desulfovibrio* (DSV) are sulfate-reducing bacteria (SRB) that are ubiquitously present in the environment and as resident commensal bacteria within the human gastrointestinal tract. Though they are minor residents of the healthy gut, DSV are opportunistic pathobionts that may overgrow in the setting of various intestinal and extra-intestinal diseases. An increasing number of studies have demonstrated a positive correlation between DSV overgrowth (bloom) and various human diseases. While the relationship between DSV bloom and disease pathology has not been clearly established, mounting evidence suggests a causal role for these bacteria in disease development. As DSV are the most predominant genera of SRB in the gut, this review summarizes current knowledge regarding the relationship between DSV and a variety of diseases. In this study, we also discuss the mechanisms by which these bacteria may contribute to disease pathology.

## 1. Introduction

*Desulfovibrio* spp. belong to the phyla Deltaproteobacteria and are mesophilic, Gram-negative, anaerobic, rod-shaped bacteria that produce hydrogen sulfide gas (H_2_S) as a terminal by-product of their metabolic activity. *Desulfovibrio* (DSV) are found in marine sediments, hydrocarbon seeps, mud volcanoes, hypersaline microbial mats, oil fields, and anaerobic waste-water treatment plants and also play a key role in the ocean sulfur cycle and in precipitation of dolomite [1,2,3]. DSV are typically motile with associated flagella, but their physiological attributes have been shown to shift depending on the surrounding environmental conditions [4]. 

*Desulfovibrio* spp. utilize the dissimilatory sulfate reduction (DSR) pathway for energy conversion by using hydrogen (H_2_) or organic compounds to reduce sulfate or oxidized sulfur compounds resulting in the production of hydrogen sulfide (H_2_S) [5,6], which can be both beneficial and detrimental to the surrounding environment. In industrial applications, DSV may contribute to corrosion of iron and steel surfaces [1,7] in anoxic conditions, presenting a challenge to companies that utilize these metals in their processes. It can also be environmentally detrimental through methylation of mercury in soil, resulting in a toxic compound [8]. On the other hand, DSV also reduce heavy metal sulfates that are soluble so that the heavy metals precipitate out of solution, enabling their recycling [1]. Other beneficial biotechnological applications include the remediation of petroleum by-products through oxidation of benzene, toluene, ethylbenzenes, and xylene [8].

More recently, DSV have become more recognized for their potential role in human diseases. While other gastrointestinal bacteria such as *f. nucleatum* and *E. coli* generate H_2_S through the assimilatory sulfate reduction pathway, the final product for this pathway is cysteine [9]. In contrast, *Desulfovibrio* are one of only five SRB, along with *Desulfobacter*, *Desulfomonas*, *Desulfobulbus*, and *Desulfotomaculum*, that use the dissimilatory sulfate reduction pathway with H_2_S as the terminal product and are minor members of the gastrointestinal microbiome [8]. Within a healthy human host, DSV localize to the pH neutral, distal colon [6] where they represent ~66% of all colonic SRB [8]. Within the mixed microbial communities of the colon, primary bacterial fermenters may rely on SRB to maintain the efficient and complete oxidation of substrates through their consumption of H_2_ produced during fermentation [6], again resulting in the production of H_2_S. In anaerobic environments with low redox potential, other anaerobes directly compete with SRB for H_2_, including those that convert H_2_ to methane (methanogens) or acetate (acetogens), with the presence of sulfate being crucial to this competition [1]. Similar to the findings for DSV in the environment, the bacterial products these organisms create can be both beneficial [10,11,12] and detrimental. In this study, we will discuss the potential role of DSV in a variety of human diseases.

## 2. DSV and Diseases

A number of diseases are associated with DSV bloom. These include conditions that not only affect the gastrointestinal tract but also affect extra-intestinal sites.

### 2.1. Bacteremia

DSV bacteremia and infections have been documented in many case reports, suggesting the impact that these bacteria can have on disease pathophysiology and extra-intestinal dissemination. However, the mechanistic details of the role of DSV in these instances remain largely unknown. A review by Goldstein et al. [13] discussed the role of DSV bacteremia. Since then, more case reports that frame DSV as an agent of bacteremia, including in the context of sepsis [14,15,16], renal cyst infection [17], bacteremia [18,19,20] thoracic endovascular aortic repair (TEVAR) [21], acute sigmoiditis [22], liver abscess [23], acute cerebral infarction [24], in choledocholithiasis and endoscopic retrograde cholangiopancreatography [25], and ulcerative colitis, have been published [26]. Among these reports, *D. desulfuricans* and *D. fairfieldensis* are usually described as the predominant species responsible for infections. However, the mechanisms underlying the pathgoenic potential of these two species remain largely unknown. As such, in case of *D. desulfuricans*, some studies have reported inflammatory effects of lipopolysaccharide (LPS) isolated from these bacteria [27,28,29]. In the case of *D. fairfieldensis,* the outer membrane vesicles (OMV) secreted by these bacteria appear to play a causal role in inflammation and tight junction barrier dysfunction [30]. Studies comparing different DSV species and their effects on the host cellular mechanisms will be important in identifying crucial targets that may be specific to each species.

### 2.2. Intestinal Bacterial Overgrowth

The number of studies reporting intestinal and extra-intestinal diseases associated with DSV overgrowth is growing. These observations suggest a more central role of these bacteria in the pathogenesis of human diseases. The identification and enumeration of DSV largely occur through sampling of feces and intestinal biopsies with subsequent bacterial culturing and real-time polymerase chain reaction (qPCR). However, each study has its own strengths and limitations. The exact relationship between increased DSV abundance and disease remains unclear and sometimes contradictory, perhaps due to the differences in sample types and methodologies. However, the majority of evidence points toward a positive association between DSV overgrowth and disease pathology and warrants further investigation into the specific role of these bacteria. Below, we discuss intestinal and extra-intestinal disorders that are linked to DSV overgrowth and include a summary of these diseases (Figure 1). Table 1 summarizes research studies that have reported the effects of DSV in various animal models in the context of these diseases.

#### 2.2.1. Inflammatory Bowel Disease (IBD)

Inflammatory bowel disease, including Crohn’s disease (CD) and ulcerative colitis (UC), is characterized by chronic inflammation of the gastrointestinal tract [31]. It is widely known that IBD patients have dysbiosis (characterized by an imbalance in the gut microbiota) [32]. Along these lines, multiple studies have identified that *Desulfovibrio* spp. are enriched in IBD. Colon mucosal biopsy samples obtained from UC patients revealed a significant increase in DSV in acute and chronic UC when compared to healthy controls [33]. In a separate study, a higher number of DSV, specifically *D. piger*, were isolated from feces of patients with IBD compared to healthy subjects [34]. A longitudinal study of UC patients conducted over a period of 1 year revealed a positive correlation between DSV with high clinical activity indices (CAI) and worse sigmoidoscopy scores (SS) [35]. In UC patients, DSV abundance was not only correlated with intestinal disease but also with anxiety and depression [36]. Adverse early-life events may also lead to the development of IBD later in life. This is exemplified by an animal study wherein prenatal maternal stress led to worse colitis in the offspring, which was also associated with DSV bloom in the offspring [37]. Interventions used to decrease the severity of IBD also reduced DSV bloom [38,39,40,41,42,43,44]. Similarly, use of probiotics in the context of disease has a protective effect that is likely attributable to their competition or inhibition of DSV [45,46]. While a causal relationship has not been clarified, there are strong indications that DSV may play a key role in IBD pathology. In contrast, some studies contradict the involvement of DSV in IBD [47,48]. 

It was also found that the small intestinal transit time was prolonged in IBD patients [49], suggesting that intestinal transit may be dysregulated in IBD patients. Moreover, an increase in the production of H_2_S has also been linked to IBD [50]. The role of DSV in causing slowing of intestinal transit was demonstrated in our previous study [51]. In this study, oral gavage of live but not heat-killed *D. vulgaris* caused the slowing of small intestinal transit in mice and this correlated with an increased concentration of H_2_S in luminal contents of the small intestine. Slowing of transit by DSV was inhibited in the presence of bismuth, a H_2_S binding compound, suggesting that H_2_S (produced by DSV) was responsible for the slower transit in these animals. These findings provide insights into the underlying mechanisms of IBD, DSV bloom, and slow intestinal transit, as well as the links between them.

The host mechanisms by which DSV may affect inflammatory outcomes have been reported in other studies. It has been demonstrated that SRB-colonized mice have increased numbers of CD11b+, B, CD8+ T, and regulatory T cells (Treg-cells) in the mesenteric lymph nodes. Colonization by SRB has also been shown to induce Th17 and Treg immune responses in mesenteric lymph nodes in germ-free mice [52]. Infection with *D. indonesiensis* has also been shown to exacerbate colitis-induced inflammation in mice [52]. In another study, oral gavage with *D. desulfuricans* aggravated atherosclerotic lesions and increased both intestinal permeability and systemic inflammation in Atherosclerosis-prone apolipoprotein E-deficient (Apoe^−/−^)-mice [53].

#### 2.2.2. Neurodegenerative Diseases

DSV bloom has also been linked to extra-intestinal disorders such as Parkinson’s disease (PD). PD is a neurodegenerative disease that mainly affects the elderly and is characterized by the loss of dopaminergic neurons in substantia nigra and by the presence of aggregates of alpha-synuclein (α-syn) protein within inclusion bodies (referred to as Lewy bodies) [54]. The cause of PD is not clear, and while PD is a neuronal disease, it has recently been proposed that its pathology may originate in the gut and that it is a bacterial-driven disease [55]. The gut microbial profile in PD patients is significantly different from that of control subjects, as evidenced by an increase in DSV in PD [56]. A recent study also demonstrated that DSV isolates from PD patients cause α-syn aggregation in *C. elegans*, suggesting a causal role for DSV in the development of PD [57]. Furthermore, a comparison between DSV from PD patients and healthy controls (patient’s respective spouses) showed that isolates from PD patients were more potent than similar *Desulfovibrio* spp. isolates from healthy controls with respect to causing substantially higher levels of α-syn aggregation. These data suggest that some factor(s) in the PD gastrointestinal environment may change certain *Desulfovibrio* spp. to make them more pathogenic. Understanding the biochemical and structural differences between DSV isolated from PD patients and those from control subjects may provide further insight into the pathogenic potential of these bacteria.

Research on mechanistic description of the association between DSV and PD suggests that a putative link may be hydrogen sulfide (H_2_S), a gaseous metabolic by-product of DSV [58]. This hypothesis was proposed in a review by Murros et al., which discussed the potential role of gut bacterial H_2_S in the development of PD via its release of cytochrome C from mitochondria, which further induces α-syn oligomerization [59]. Cytochorme C is capable of possessing peroxidase activity via its interaction with anionic lipids that utilize α-syn as a substrate. This leads to oligomerization of syn with cytochrome C into high molecular weight aggregates [59,60]. Alpha-syn aggregates constitute major components of Lewy bodies, a hallmark of PD [61]. Hydrogen sulfide not only impacts mitochondrial function but also reduces ferric iron to ferrous iron, and ferrous iron may promote α-syn aggregation [62,63,64,65,66,67]. A computational metabolomic profile of PD patients revealed significant changes in the pathways related to gut microbial sulfur metabolism, supporting the idea that an increase in DSV abundance may lead to increased exposure to H_2_S (through enhanced sulfur metabolism) [68]. Whether or not DSV-derived H_2_S mediates the formation of α-syn aggregates remains to be investigated. In contrast to these findings, H_2_S has been shown to have protective effects on mitochondria function in hyperhomocysteinemia, which is associated with the development of neurological disorders including Parkinson’s disease and Alzhiemer’s disease (AD) [69,70,71].

Hydrogen sulfide is not the only DSV by-product with effects in PD. Bacterial endotoxin lipopolysaccharide (LPS) may also influence PD outcomes. In an animal model of rotenone-induced PD, DSV overgrowth occurred in PD mice, but fecal microbial transfer (FMT) from healthy mice reversed this phenomenon. Suppression of DSV overgrowth occurred when LPS-triggered Toll-like receptor 4 (TLR4) inflammatory pathway was inhibited [72]. 

Since DSV are positively correlated with PD and appears to contribute to the development of the disease by causing α-syn aggregation, it is possible that DSV also contribute to other neurodegenerative diseases associated with protein aggregation and inclusion bodies such as those found in Huntington’s disease (HD), Alzheimer’s (AD), and amyotrophic lateral sclerosis (ALS). Possibly, its role in these diverse neurodegenerative conditions may depend on a common mechanism. While it is not clear whether or not all these neurodegenerative conditions are associated with DSV bloom, they are universally associated with gut microbial dysbiosis [73,74,75].

#### 2.2.3. Autism

Autism spectrum disorders (ASD) are neurobiological disorders that cause impaired social and communication skills and are characterized by repetitive behaviors (https://www-cdc-gov.libproxy.unm.edu/ncbddd/autism/facts.html, accessed on 15 May 2023). Many studies have reported an increase in the abundance of DSV in context of autism. For instance, it was found that children with ASD had more DSV when compared to healthy children [76,77,78,79]. Additionally, severity of ASD appears to be proportional to DSV abundance [80]. Despite this strong correlation, studies aiming to understand the mechanistic link between DSV overgrowth and ASD are lacking. Some studies have suggested a few mechanisms by which DSV may be responsible for ASD. One such study by Karnachuk et al. identified low iron bioavailability caused by DSV as a result of its H_2_S production [81]. By binding iron, hydrogen sulfide forms iron sulfide, resulting in iron deficiency, a feature of ASD [82,83]. Another mechanism by which DSV may contribute to ASD is through their production of LPS endotoxin. This is supported by findings that show increased endotoxin concentrations in the serum of ASD patients when compared to healthy controls [84]. LPS may contribute in ASD by causing neuroinflammation [85], which has been implicated in the pathogenesis of ASD [86]. Another mechanism that has been proposed to explain the effect of DSV is their production of a short chain fatty acid, propionic acid, as increased propionic acid is also found in autism cases [87,88,89,90,91]. Propionic acid infusion was found to induce repetitive behaviors and other effects associated with ASD in rodents [88,92]. Propionic acid was also found to alter the phospholipid profiles in these animals. It has been suggested that ASD may be associated with dysregulated fatty acid metabolism in ASD [93], thus indicating the underlying mechanism of propionic acid-induced ASD-like behavior. Not all studies point toward a pathological role for DSV in ASD. It has been shown that microbial transfer therapy (MTT) improved the symptoms of autism and increased DSV [94]. Future studies are needed to clarify the relationship between ASD and DSV abundance and to identify underlying molecular mechanisms behind their association.

#### 2.2.4. Cognition

A role for DSV in impairing cognition has been demonstrated in our laboratory [95]. Oral gavage of live *D. vulgaris* was found to impair radial arm maze performance and Morris water maze performance in mice in tests used to assess spatial learning and memory. This effect was not observed in animals gavaged with heat-killed DSV, suggesting that metabolically active DSV were required for this effect. Interestingly, the concentration of H_2_S was also higher in the small intestine and cecum of mice gavaged with live but not heat-killed DSV. Thus, production of H_2_S by DSV may be responsible for causing learning and memory defects in mice gavaged with DSV. While the physiological levels of H_2_S are important in many neuronal functions, including learning and memory, elevated levels of H_2_S are toxic and have adverse effects on many physiological functions including poor memory and other neurobehavioral deficits [96,97,98,99,100]. Understanding the mechanism of how H_2_S generated by DSV in the gut may cause memory deficits may provide a crucial link to understanding the basis of neuronal diseases related to impaired cognitive memory function and associated with DSV bloom.

#### 2.2.5. Cancer

Several studies have reported an increase in *Desulfovibrio* genera in cancer patient samples when compared to healthy control groups. This has been observed in cases of rectal cancer [101], adenomatous polyps [102], colorectal cancer, gastric cancer, breast cancer, and lynch syndrome. A study using logistic regression analysis reported salivary *D. desulfuricans* to be a predictor of risk for colorectal cancer (CRC) [103]. Role of DSV in the development of CRC was also studied by administering DSV via oral gavage in rodents [104]. DSV caused colonic damage and induced a pre-metastatic niche (PMN) in the liver by increasing Matrix metalloproteinase 2 (MMP2), Matrix metalloproteinase (MMP9), and C-X-C motif chemokine ligand 12 (CXCL12), components of extracellular matrix that are important in the formation of PMN. PMNs are microenvironments that make secondary organs conducive to tumor metastasis [105]. The main player in DSV-induced cancer phenotypes appears to be the H_2_S produced by these bacteria. H_2_S is known to cause DNA damage and contributes to CRC [106]. The role of H_2_S in mitochondria has also been discussed in context of cancer [107,108]. The abundance of DSV in established animal models of cancer has also been tested. In animal models of 1,2-dimethylhydrazine (DMH)-induced colon cancer, an increase in DSV density was observed in the tumor group when compared to the control group [109]. A sequential increase in DSV also occurred in mice that developed Non-Alcoholic Fatty Liver Disease (NAFLD)-associated hepatocellular carcinoma (NAFLD-HCC) in response to dietary cholesterol [110]. In contrast to these studies, no difference was found in DSV density in either patients with colorectal cancer, patients with upper gastrointestinal cancer, or healthy controls [111]. Similarly, no difference was found in DSV density in polypectomized (PP) individuals and healthy individuals, while DSV numbers were reduced in CRC individuals when compared to the PP and healthy groups [112]. In a study assessing post-operative pain in women with breast cancer, DSV was negatively associated with sleep disturbance and anxiety [113].

Various therapies aimed at treating human cancer patients and animal models have shown that DSV abundance can be successfully suppressed with probiotic treatments and that such treatments could improve clinical outcomes. For example, supplementation with the probiotic *C. butyricum* significantly reduced the abundance of DSV in gastric cancer patients after gastrectomy [114]. In another study, oral administration of *Bifidobacterium* reduced the abundance of DSV and improved dextran sodium sulfate (DSS)-induced colitis and colon cancer in rats [41]. Similarly, *Lactobacillus coryniformis* MXJ32 administration decreased DSV and ameliorated azoxymethane/DSS-induced colitis-associated colorectal cancer, enhanced tight junction proteins, and downregulated proinflammatory cytokines [115].

#### 2.2.6. Metabolic Syndrome

Many studies have reported a positive correlation between DSV and obesity and other metabolic syndrome phenotypes. A high abundance of DSV was observed in obese patients with Prader–Willi syndrome, a genetic mutation that often leads to obesity [116]. Metagenomic and other profiling studies have revealed an abundance of DSV in Type 2 diabetes (T2D) [117,118,119]. An increase in DSV was also observed in coronary artery disease (CAD) patients with Type-2 diabetes mellitus (DM2) when compared to CAD patients without DM2 [120]. Upon examining the gut microbiota of women with gestational diabetes mellitus (GDM), it was found that DSV load was higher in GDM in comparison to healthy pregnant women [121]. Various studies have reported the reversal of DSV abundance in T2D models by different compounds found in natural medicines and extracts derived from plants. For example, a high-fiber diet ameliorated T2D serological and psychiatric outcomes in a randomized control study, with these outcomes correlating with a decrease in DSV abundance [122]. 

The protective effects of probiotics have been tested in improving diabetes-related outcomes. It was found that orally administered *L. acidophilus* improved epithelial barrier function; lowered proinflammatory cytokines such as interleukin-8 (IL-8), tumor necrosis factor-α (TNF-α), and interleukin-1β (IL-1β) in liver and colon tissue; and prevented liver and colon tissue injuries to some extent. At gut microbial level, *L. acidophillus* decreased the abundance of DSV [123]. Additionally, both diet (consisting of pre- and pro-biotics as well as whole grains) and FMT have been shown to control blood glucose and blood pressure levels in T2D patients while also significantly decreasing DSV and other SRB [124]. In another study, consumption of high dietary fiber reduced T2D and DSV in Chinese patients [125]. In other interventional studies in pre-clinical models, various compounds have been shown to have a protective effect against T2D and to decrease the DSV load [126]. 

While most reports have described a positive association between DSV and disease, some studies have reported the opposite. One such study highlighted the protective role of DSV in NAFLD via its production of acetic acid [127]. In a recent study, supplementation with a slowly digestible carbohydrate diet improved hyperglycemia and hyperlipidemia in high-fat diet-streptozocin-induced diabetic mice, resulting in an increase in DSV population [128]. However, the role of DSV in these studies is not clear, and further research demonstrating a direct role of DSV in the development of metabolic syndrome is needed. 

Diet also plays an important in the development of metabolic syndrome. As such, DSV bloom has been observed in High-Fat Diet (HFD)-fed animal models, suggesting a positive correlation between DSV and Western diet [104]. A study demonstrated that maternal HFD-induced gut microbiota disturbance in offspring at weaning included an increase in DSV [129], which also occurred in conjunction with increase in body weight, hyperglycemia, glucose intolerance, hyperinsulinemia, hypercholesterolemia, and leptin resistance, and a decrease in adiponectin in the offspring. It was also reported that HFD caused anxiety-like behavior in addition to disrupted biochemical markers in obesity prone mice but not in the obesity resistance group [130]. This was correlated with increased DSV in the small intestines of these animals. A more direct role for DSV in HFD-induced outcomes was established by gavaging *Desulfovibrio piger* in HFD-fed mice. *D. piger* increased hepatic steatosis and fibrosis in HFD-fed mice [131]. *D. piger* also caused increased intestinal permeability in mice fed with HFD by disrupting the tight junction protein ZO-1. Many studies have reported amelioration of HFD-induced non-alcoholic fatty liver diseases (NAFLD) as well as other physiological and biochemical changes in rodents by various nutritional supplements and other compounds such as plant polyphenols. These changes were correlated with a reduction in HFD-induced abundance of DSV and other gut bacteria [132,133,134,135,136,137,138,139,140,141]. A few studies have also revealed the protective effect of probiotics such as Lactobacilli and Bifidobacteria in ameliorating HFD-induced changes along with decreasing HFD-induced increase in DSV population [142]. Additionally, it was found that exercise improved Western diet (WD)-induced atherosclerosis by modulating gut microbiota, including WD-induced increase in DSV. On the other hand, certain environmental and man-made pollutants can exacerbate the risk of HFD-induced metabolic disorders, which is also correlated with exacerbated HFD-induced DSV bloom [143]. A high-fat, high-fructose corn syrup-based, high-cholesterol western-style diet was found to cause obesity, dyslipidemia, and systemic insulin resistance in juvenile Ossabaw swine pigs compared to control pigs. At gut microbial level, these pigs also had increased cecal DSV [144]. In contrast to HFD, a diet rich in fiber showed a negative association with DSV abundance. In a cross-sectional and a longitudinal study involving Chinese patients [125], it was shown that high fiber intake among diabetes patients lowered the abundance of DSV. Understanding the mechanistic phenomena underlying positive association between DSV and HFD-induced changes may be helpful in understanding the development of diet-related conditions such as obesity. 

**Table 1 microorganisms-11-01772-t001:** In vivo effects of DSV in various animal models.

Disease Context	System	DSV Source	Methods	Findings	Reference
Parkinson’s Disease	*C. elegans*	DSV isolates from feces of PD patients	Worms expressing α–syn-YFP fed on DSV containing medium	Alpha synuclein aggregation in brain	[57]
Inflammatory Bowel Disease	C57/BL6 mice	*D. vulgaris* Hildenborough (ATCC	Intestinal transit of a fluorescent probe through the small intestine in animals gavaged with DSV	Intestinal transit was slowed down	[51]
Experimental Colitis	C57BL/6 mice (germ free and wild type)	*D. indonesiensis* isolated from biofilm on corroded ship or human SRB consortium from patients with colitis	H&E staining Cytokine analysis by CBA Th1/Th2/Th17 kit	H&E staining Cytokine analysis by CBA Th1/Th2/Th17 kit	[52]
Atherosclerosis	C57/Bl6 Apoe^−/−^ Caco2	*D. desulfuricans*	Intestinal permeability using FITC probe Inflammatory markers and tight junction proteins	Increased formation of atherosclerotic lesion Increased inflammation Increased intestinal permeability	[53]
Colorectal Cancer	BALB/c Mice	*Desulfovibrio* (species unspecified), from China General Microbiological Culture Collection Center	Real-Time qPCR ELISA kits for LPS, H_2_S.	Decreased mRNA for tight junction proteins Increased mRNA for inflammatory markers Increased mRNA levels of extacellular matrix proteins important for formation of pre-metastatic niche (PMN) in the liver Increased serum ALT and AST Increased H_2_S and LPS in serum	[109]
Obesity	C57Bl/6 Myd88^LoxP/LoxP^ mice crossed to C57Bl/6 CD4-Cre animals to produceCD4-Cre^+^ (T-MyD88^−/−^) animals	*D. desulfuricans* subsp. *desulfuricans*	qPCR 16s rDNA sequencing	Expression of CD63 was increased by DSV Reduction in Clostridia	[145]
Cognition	C57/BL6 mice	*D. vulgaris* Hildenborough (ATCC)	8-Arm radial learning maze performance Morris water maze performance H_2_S measurement	Impaired working memory Increased H_2_S in small intestine and cecum	[95]

Role of DSV in animals that had T cell specific ablation of the innate adaptor molecule, Myeloid differentiation factor 88 (Myd88) (T-Myd88^−/−^ mice), which causes obesity in mice, was also addressed [145]. These animals develop dysbiosis mirroring that of individuals with obesity. Firstly, T-Myd88^−/−^ mice had a higher load of DSV and a loss of beneficial *Clostridia* (related to leanness) when compared to wild-type mice. Additionally, colonization of germ-free animals with *D. desulfuricans* led to a significant reduction in *Clostridia*. CD36 is a regulator of lipid absorption in the small intestine [146,147]. DSV elevated the expression of CD36, thus contributing to dysfunctional absorption of lipids, leading to obesity phenotypes. 

#### 2.2.7. Other Diseases

Several other diseases have also been linked to DSV bloom. Prevalence of DSV was found to be significantly higher amongst orthodontic patients when compared to non-orthodontic participants (about 20%) [148]. DSV was found to be abundant in patients with periodontitis [149]. Other studies have reported an increased abundance of DSV associated with diseases such as Chronic Kidney Disease (CKD) [150,151]. DSV has also been found to be associated with liver cirrhosis [152], cute myocardial infarction (AMI) [153], stroke, and transient ischemic attack patients [154]. DSV is also found to be more abundant in patients with irritable bowel syndrome (IBS) [155,156] and those with autoimmune diseases such as systemic sclerosis [157] and multiple sclerosis [158].

Overall, majority of studies in the literature support a positive relationship between DSV and various diseases. However, detailed investigations that take into account the discrepancies observed among different studies are needed to gain more clarity on the nature of these associations. 

## 3. Effect of DSV on Host Cells In Vitro

While the effects of DSV have been studied in vivo by inoculating either germ-free mice or conventionally raised mice with monocultures of DSV species or with SRB consortia, direct mechanistic cellular events affected by DSV have been most effectively studied in vitro using various cell lines such as macrophages and epithelial cell lines (Table 2).

It has been shown that enriched SRB culture from UC patients, but not those from control subjects, induced apoptosis of epithelial cells [159]. In addition, pure culture of *D. indonesiensis* was internalized and induced apoptosis in human ileocecal adenocarcinoma HCT 8 cells. This effect was exacerbated when a co-culture of DSV with *E. coli* 2R/BP was used. Furthermore, antibody raised against the exopolysaccharide (EPS) of *D. indonesiensis* cross-reacted with SRB population from UC patients, but not with SRB combination from non-UC controls, suggesting that antibodies raised against the EPS of *D. indonesiensis* could be utilized as a marker to differentiate between SRB from UC versus non-UC biopsies.

As mentioned previously, role of SRB in periodontitis has been described in the literature. Different SRB have also been isolated from the oral cavity, generally belonging to the genus *Desulfovibrio* [160,161,162,163,164,165]. Proinflammatory effects of *D. desulfuricans* and *D. fairfieldensis* on human KB oral epithelial cells have also been evaluated elsewhere [166]. Both strains invaded KB cells and induced IL-6 and IL-8 cytokine expression. Similarly, DSV have been shown to induce IL-6 and IL-8 production in human gingival fibroblast cells HGF-1 [29].

Role of DSV in activating cellular pathways has also been reported, including in our own study, which showed that the *D. vulgaris* -induced infection of RAW264.7 macrophage-like cell lines activated proinflammatory Notch cell-to-cell signaling pathway and induced the protein expression of proinflammatory cytokine IL-1β [167]. DSV have also been demonstrated to induce nitric oxide (NO) in RAW 264.7 macrophage-like cell lines, which was inhibited in the presence of N(G)-monomethyl L-arginine (L-NMMA), a NO synthase inhibitor [168]. Additionally, DSV impair intestinal barrier integrity. Work from our laboratory showed that *D. vulgaris* increased barrier permeability in polarized intestinal Caco-2 cells by disrupting the localization of an important tight junction protein occludin [169], and DSV mediated this effect via upregulating Snail1 transcription factor. DSV may also compromise intestinal barrier integrity indirectly by inhibiting lysozyme [170], an anti-microbial protein crucial in shaping the gut microbiota [171].

**Table 2 microorganisms-11-01772-t002:** In vitro effects of DSV on cell lines.

Cell Lines	DSV Source	Methods	Findings	Ref.
HCT116	D. indonesiensis mono-culture or co-culture with E. coli isolate 2R/BP SRB consortia from human biopsy samples	Flow cytometry and tunnel labeling for apoptosis Immunostaining	Induction of apoptosis antibody against exopolysaccharides of D. indonesiensis cross reacted with the SRB from UC patients but not with the SRB from non-UC controls.	[159]
KB cell line ATCC CCL-17	D. desulfuricans ATCC 29577, D. desulfuricans ATCC 27774, D. fairfieldensis ATCC 700045	Invasion assay Electron microscopy ELISA kit for cytokines	DSV invaded KB cells in microtubule dependent manner DSV are present in the free space in cytoplasm Induction of pro-inflammatory cytokines by DSV	[166]
RAW 264.7	*Desulfovibrio vulgaris* Hildenborough (ATCC 29579)	Western blot siRNA transfection qPCR	Increased mRNA and protein expression of Notch1 and IL-1b. Activation of Notch intracellular domain Paracrine activation of Notch signaling in recipient cells by soluble factors in culture supernatant of DSV-treated cell.	[167]
RAW 264.7	*Desulfovibrio vulgaris* Hildenborough (ATCC 29579)	Colorimetric assay for nitrite production	Increased nitrite production in *D. vulgaris*-infected macrophages	[168]
Polarized and differentiated Caco2	*Desulfovibrio vulgaris* Hildenborough (ATCC 29579)	FITC flux to measure barrier permeability siRNA transfection Western blot	Increased paracellular permeability Increased snail protein expression	[169]
RAW 264.7	*Desulfovibrio vulgaris* Hildenborough (ATCC 29579)	Western blot Lysozyme activity assay	Decreased lysozyme mRNA and protein expression	[170]

## 4. Products of DSV Responsible for Causing Potentially Harmful Effects

### 4.1. Hydrogen Sulfide

Hydrogen sulfide is a by-product of *Desulfovibrio* dissimilatory reduction pathway and is by far the most important player in imparting pathogenicity to these bacteria. Involvement of DSV-derived H_2_S has been discussed throughout this review in the context of different diseases. We have covered the effects of mammalian and bacterial H_2_S on host pathophysiology elsewhere [96].

To generate H_2_S, DSV use lactate and hydrogen (H_2_) as electron donors and sulfate as the terminal electron acceptor [1,8,9]. This process requires transfer of eight electrons and processing of enzymes in both the cytoplasm and periplasm of the bacteria, with breakdown of lactate resulting in acetate production. Sulfate and sulfite come from dietary sources such as food preservatives and antioxidants or sulfate is released from its bound form in mucin or chondroitin by saccharolytic bacteria such as *Bacteroidetes* spp. [6,9]. *Desulfovibrio* adenosine-5′-phosphosulfate (APS) sulfurylase binds ATP to the sulfate, resulting in the production of two inorganic phosphates and APS. Cytoplasmic APS reduction is catalyzed by APS reductase, where two electrons are used to make adenosine monophosphate (AMP) and sulfite is an interval product. Cytoplasmic sulfite is then further reduced to hydrogen sulfide by dissimilatory sulfite reductase (desulfoviridin in *Desulfovibrio*) (Figure 2), which is then released into the environment [1,8,9]. The toxicity of hydrogen sulfide is dependent on its oxidation state (H_2_S vs. HS^−^ vs. S^2−^), which in turn is dependent on pH of the environment [172]. Healthy levels of H_2_S fall into a range of 0.3–3.4 mmol/L in the intestine [173], with higher concentrations resulting in not only the destruction of colonocytes but also in attenuation of bacterial ribosomal proteins and of the genes encoding the dissimilatory enzymes (>25 mM H_2_S) [174]. High H_2_S concentrations also cause an increase in the expression of bacterial genes involved in proteolysis [175]. Hydrogen sulfide can be beneficial to the system in low concentrations, but at higher concentrations, it is toxic to butyrate fermentation by colonocytes (which is responsible for 70% of the energy required by colonocytes) andto *Desulfovibrio* as well [176], although *Desulfovibrio* are capable of tolerating much higher concentrations of hydrogen sulfide (>10 mM [174]).

While the effects of H_2_S in causing cellular and physiologic damage are well known, studies demonstrating direct effects of DSV-produced H_2_S on host cells are scarce. Conducting investigations studying the role of DSV-derived H_2_S in pathogenesis of DSV-associated diseases will be key to identifying the underlying mechanisms of DSV-induced effects on the host.

### 4.2. LPS

Lipopolysaccharide is an important endotoxin produced by Gram-negative bacteria that is responsible for a plethora of harmful biological effects. The chemical composition of *D. desulfuricans* LPS was studied by Lodowska et al. [177]. LPS typically consist of lipid A, the core oligosaccharide, and an O-specific polysaccharide called the O antigen. Lipid A structure in *Desulfovibrionaceae* was also characterized in another study [178]. In this study, LPS from two SRB isolates with differing lipid A moieties in the LPS were isolated from the same healthy human gut. These different LPS were then tested on human THP-1 cells and their proinflammatory effects via the production of inflammatory cytokines IL-6 and TNFα, were observed. SRB1 LPS was found to be less potent for cytokine production than SRB2 LPS, suggesting that SRB may differ in their pathogenic potential due to the differential proinflammatory potential of their LPS. In another study, LPS isolated from *D. desulfuricans* was found to cause an increase the production of pro-inflammatory cytokines IL-6 and IL-8 while also inducing the expression of E-selectin and VCAM-1 in endothelial cells [27].

*D. desulfuricans* LPS also increased the secretion of IL-6 and IL-8 in gingival fibroblasts [29] but failed to elicit IL-8 production in Caco-2 epithelial cells [179]. It is possible that since DSV are normal residents of colon, its LPS may not be immunostimulatory for colonic cells. In another study, the effects of LPS from either *D. desulfuricans* intestinal isolates or soil isolates on cytokine secretion in Caco-2 were characterized [180]. LPS derived from the soil isolates was more potent for IL-8 secretion than that derived from the intestinal isolates. In contrast, IL-6 induction was much higher in response to LPS from the intestinal strain compared to LPS from the soil strain. However, in a separate study, LPS derived from *D. desulfuricans* was found to elicit the gene expression of NFκβp65 subunit as well as IκB gene expression in Caco-2 cells [28]. Whether changes in gene expression led to the activation and localization of NFκβ and whether there was an effect on cytokine production, was not addressed in this study. LPS from *D. desulfuricans* was also found to cause secretion of proinflammatory cytokine IL-6 as well as the neutrophil-, basophil-, and T-cell-attracting chemokine IL-8 in human gingival fibroblast (HGF-1) cell line [29] and in human umbilical vein endothelial (HUVEC) cells [27]; IL-6 secretion was also induced in Tamm–Horsfall protein 1 (THP1) cells [178]. Together, these results suggest that different pathogenic effects of DSV-derived LPS may not only be dictated by the type of DSV strains but may also depend on both the discrimination of the endotoxin type by epithelial cells and the particular cell type (epithelial, macrophage, endothelial, etc.) involved in the response.

### 4.3. Extracellular Vesicles

Outer membrane vesicles (OMVs) are double-layer lipid membrane nanospheres that are commonly produced by Gram-negative bacteria that range in size from 20 to 300 nm [181,182]. OMVs play an important role in bacterial physiology as well as in stress response. OMVs are one of the means by which bacteria effectively communicate with their environment [183,184].

Recent studies have identified outer membrane vesicles (OMVs) that are secreted by *D. fairfieldensis.* This bacterium is potentially the most pathogenic strain of DSV that has been identified thus far [185] and is associated with Choledocholithiasis and Endoscopic Retrograde Cholangiopancreatography [25]. OMVs isolated from *D. fairfieldensis* are responsible for inducing inflammation and pyroptosis in macrophages. OMVs are known to activate immune responses [181] and stimulate the production of proinflammatory cytokines such as IL-1β, IL-8, and TNFα [186,187]. OMVs have also been reported to disrupt the epithelial barrier junction. OMVs derived from *D. fairfieldensis* disrupted epithelial tight junction in Caco-2 cells by downregulating ZO-1 and the occludin gene and protein expression [30] in a manner similar to that observed in *C. jejuni* infection, where OMVs cleave E-cadherin and occludin, important players in maintaining tight junctions [188]. A study also purified and identified outer membrane-associated proteins in *D.*
*vulgaris* [189]. Thus, OMVs secreted by DSV play an important role in the pathogenic potential of this bacteria. Whether or not other species of *Desulfovibrio* produce OMVs and whether OMVs produced by DSV can be targeted for therapy should be explored in future research.

### 4.4. Mucolytic Activity of DSV

There are studies that have reported the colonic mucin binding of DSV in UC. It has been reported that mucin binding profiles of clinical isolates of *Desulfovibrio* spp. were specific to each isolate [190]. The mucus gel layer is a physical barrier that is important in maintaining intestinal barrier integrity. Increased microbial colonization of mucus has been observed in IBD [191,192,193]. SRBs metabolize sulfated mucopolysaccharides found in mucin [194]. It was found that there was an inverse correlation between sulfated mucin and DSV and inflammation in UC [191]. Thus, DSV may contribute in the pathogenesis of IBD by depleting the protective mucus layer, which further exacerbates outcomes such as intestinal barrier loss, increased inflammation, and other hallmarks of IBD. Figure 3 summarizes the DSV products that may be responsible for the pathogenic potential of DSV.

## 5. Conclusions

DSV bacteria have emerged as important pathobionts in the last few years and appear to be contributing factors in not only intestinal disorders but also in extra-intestinal diseases. This is exemplified by diseases such as Parkinson’s disease, where DSV appear to play a causal role in the formation of synuclein aggregates, a hallmark of PD. It also provokes the question of whether the association and involvement of DSV can be extended to similar diseases such as ALS, HT, or AD, where DSV may act as a common denominator in disease pathophysiology. Similarly, role of DSV in the development of atherosclerosis has been reported. Thus, mounting evidence suggests that DSV have multifaceted effects throughout the human body, extending beyond their influence in the gut. This also applies to all other intestinal and extra-intestinal diseases that are linked to gut microbial dysbiosis where DSV overgrowth may occur. While a few studies have revealed some cellular pathways by which DSV may mediate their effects, the underlying mechanisms of how DSV may affect these diseases remain largely unknown. Thus, future in vivo and in vitro studies should focus on determining the molecular mechanisms behind the effects of DSV on the host. As some DSV species appear to be more pathologically significant than others and because DSV isolated from patients have worse outcomes on recipient cells than the ones isolated from healthy individuals, it is critical to understand the structural (such as LPS), biochemical, and functional differences between DSV species. This may hold a key to identifying the target mechanisms that are unique to each of these species. It is also important to understand what host factors in patients (but not in healthy controls) may lead to increased pathogenicity of these strains. Overall, this information may be helpful in identifying novel therapeutic targets that could be utilized in the management or treatment of diseases that are associated with DSV bloom.

## Figures and Tables

**Figure 1 microorganisms-11-01772-f001:**
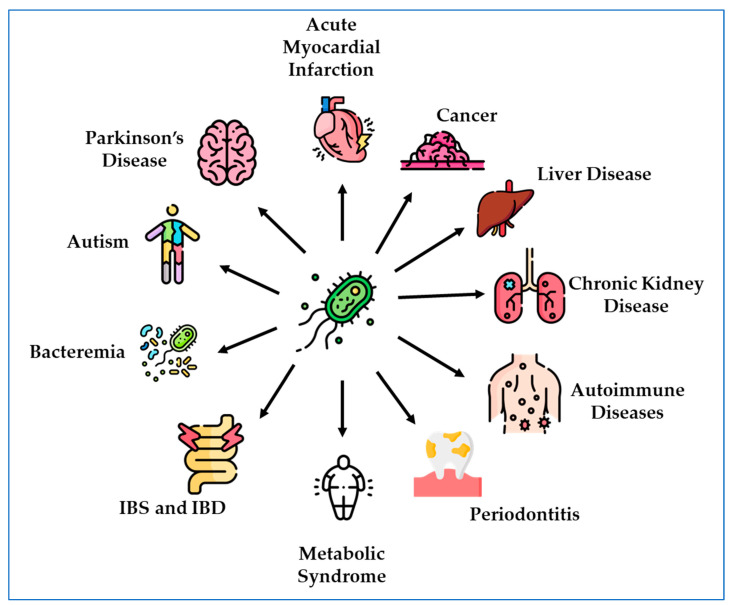
Association between DSV and various intestinal and extra-intestinal diseases.

**Figure 2 microorganisms-11-01772-f002:**
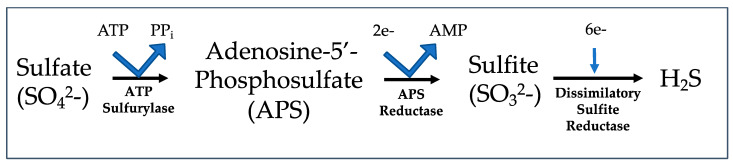
Dissmilatory sulfate reduction pathway.

**Figure 3 microorganisms-11-01772-f003:**
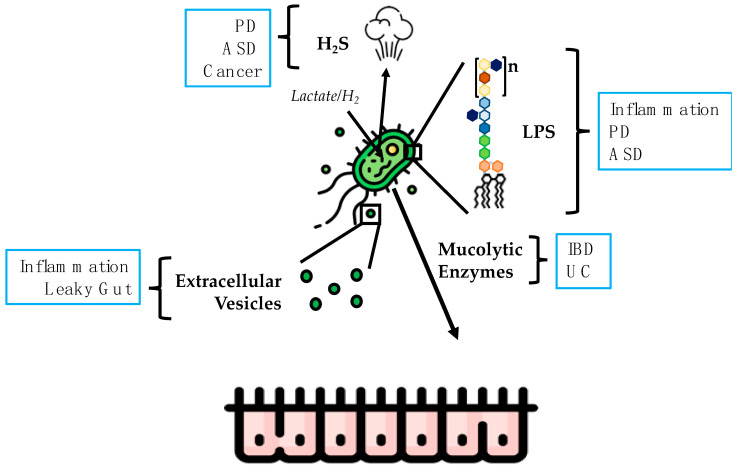
DSV products that may be responsible for imparting pathogenicity to DSV.

## Data Availability

No new data were generated or analyzed in this study. Data availibility is not applicable to this article.

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
