# Peer review of "Desulfovibrio in the Gut: The Enemy within?"

_microorganisms, 2023, doi:10.3390/microorganisms11071772_

Round 1
Reviewer 1 Report
The review provides valuable insights into Desulfovibrio (DSV), sulfate-reducing bacteria (SRB) commonly found in the environment and as commensal bacteria in the human gastrointestinal tract. While they typically exist in low numbers in a healthy gut, DSV can opportunistically overgrow in the presence of certain intestinal and extra-intestinal diseases. The review presents works that show a positive association between DSV overgrowth (a bloom) and various human diseases. Given that DSV is the predominant SRB genus in the gut, this review aims to provide a comprehensive summary of the current understanding of the relationship between DSV and various diseases.
As a reviewer, I found the review to be highly enjoyable and commend the authors for their comprehensive survey on DSV in the human body. The review provides valuable insights into DSV's presence in the gut and other areas of the human body. After carefully examining the review, I find that the title question may need further consideration. DSV appears to have multifaceted effects throughout the human body, extending beyond its influence on the gut. This observation could be further refined and emphasized in the conclusions section.
To enhance paper's organization and facilitate reader comprehension, I recommend including a well-structured table that allows for a systematic comparison of the different effects of DSV on various diseases, accompanied by supporting in vitro or in vivo evidence. Such a table would connect different parts of the article and contribute to a more cohesive reading experience. Consequently, the need for Figure 1, may be eliminated.
Additionally, it would be beneficial to explore the role of DSV in relation to lag in the breathing circuit and H2S decomposition, as mentioned in the text. Furthermore, adding a table describing the research methods employed in DSV studies would be valuable.
Regarding the paper's structure, I suggest shortening the introduction section, particularly the parts unrelated to diseases. It would be appropriate to incorporate both in vivo and in vitro evidence within the subsections dedicated to specific diseases. A separate chapter dedicated solely to evidence may not be necessary, as much of it will likely be relevant to the previously mentioned diseases.
Line 76: “D.desulfuricans and D. fairfieldensis are identified as the predominant.
Species responsible for infections. “ This is a good, and central point, but it will need to give more information and explanations on how Extensive research has attributed their infection prevalence to specific factors and mechanisms.
Minor comments:
Line 121: Change the title to “neurodegenerative diseases “ because there is a discussion on other diseases except Parkinson's.
Line 147: Is it specific to Parkinson’s Disease the “role of H2S in mitochondria “? If not, maybe it should discuss in different sections.
e
Author Response
We thank the reviewer for their valuable time and suggestions.
The review provides valuable insights into Desulfovibrio (DSV), sulfate-reducing bacteria (SRB) commonly found in the environment and as commensal bacteria in the human gastrointestinal tract. While they typically exist in low numbers in a healthy gut, DSV can opportunistically overgrow in the presence of certain intestinal and extra-intestinal diseases. The review presents works that show a positive association between DSV overgrowth (a bloom) and various human diseases. Given that DSV is the predominant SRB genus in the gut, this review aims to provide a comprehensive summary of the current understanding of the relationship between DSV and various diseases.
We thank the reviewer for their valuable comments and suggestions. We agree that all the points raised by the reviewer are important in improving the quality of this manuscript and have been addressed to the best of our ability.
As a reviewer, I found the review to be highly enjoyable and commend the authors for their comprehensive survey on DSV in the human body. The review provides valuable insights into DSV's presence in the gut and other areas of the human body. After carefully examining the review, I find that the title question may need further consideration. DSV appears to have multifaceted effects throughout the human body, extending beyond its influence on the gut. This observation could be further refined and emphasized in the conclusions section.
We have now expanded the conclusion section.
To enhance paper's organization and facilitate reader comprehension, I recommend including a well-structured table that allows for a systematic comparison of the different effects of DSV on various diseases, accompanied by supporting in vitro or in vivo evidence. Such a table would connect different parts of the article and contribute to a more cohesive reading experience. Consequently, the need for Figure 1, may be eliminated.
We have now incorporated two tables describing the direct role of DSV in various diseases and also the direct effects of DSV in cell lines.
Additionally, it would be beneficial to explore the role of DSV in relation to lag in the breathing circuit and H2S decomposition, as mentioned in the text. Furthermore, adding a table describing the research methods employed in DSV studies would be valuable.
We did not find any mention of “lag in breathing circuit and H2S decomposition” in the text. The methods employed in understanding the role of DSV studies has been included in tables1 and 2.
Regarding the paper's structure, I suggest shortening the introduction section, particularly the parts unrelated to diseases. It would be appropriate to incorporate both in vivo and in vitro evidence within the subsections dedicated to specific diseases. A separate chapter dedicated solely to evidence may not be necessary, as much of it will likely be relevant to the previously mentioned diseases.
We agree with the reviewer in staying focused on disease in the aspect of DSV in the introduction, but in order to accommodate reviewer 2 suggestions, we have kept non-diseases aspect including the industrial and biotechnological uses of DSV in the review but in a concise fashion (lines 40-50).
We have also moved the invivo data to respective disease section. However, we kept the invitro section separate as it cannot be assigned to any particular disease section.
Line 76: “D.desulfuricans and D. fairfieldensis are identified as the predominant.
Species responsible for infections. “ This is a good, and central point, but it will need to give more information and explanations on how Extensive research has attributed their infection prevalence to specific factors and mechanisms.
There is no information on the mechanism of why these strains are more prevalent in terms of infection. But, we have added some explanation on this in lines 88-95 to expand this.
Minor comments:
Line 121: Change the title to “neurodegenerative diseases “ because there is a discussion on other diseases except Parkinson's.
We have changed this (now line 153)
Line 147: Is it specific to Parkinson’s Disease the “role of H2S in mitochondria “? If not, maybe it should discuss in different sections
Role of H2S in mitochondria may not be specific to Parkinson’s Disease. We have added lines 189-192. We have moved some of this information in other sections such as cancer (line 267).

Reviewer 2 Report
The manuscript titled (Desulfovibrio in the gut, the enemy within?) by Singh et al. summarized the current knowledge on the relationship between Desulfovibrio and a variety of diseases, as well as discussing the mechanisms by which these bacteria may contribute to disease pathology. The manuscript could be accepted after covering some issues.
1- Some minor English and grammatic mistakes should corrected throughout the manuscript.
2- In introduction, please add all types of SRB.
3- What about the biotechnological uses of Desulfovibrio.
4- There are any active constituents that are isolated from this type of bacteria. If yes, please add all and draw the chemical structures.
5- Citation of references in the text is uncorrected example [13] [14] [15]...........please correct all as [13-15].
6- List of abbreviations with complete name should be provided.
7- Dear authors, any review should attract the readers by providing figures or equations for the mechanisms either for its benefits or hazards. So, please add some figures to the review to illustrate the bacterial mechanism in production of H2S, bacteremia, IBD, and others….., Just enumerating the benefits or hazards is not enough.
Moderate editing of English language required
Author Response
We thank the reviewer for their valuable time and suggestions.
The manuscript titled (Desulfovibrio in the gut, the enemy within?) by Singh et al. summarized the current knowledge on the relationship between Desulfovibrio and a variety of diseases, as well as discussing the mechanisms by which these bacteria may contribute to disease pathology. The manuscript could be accepted after covering some issues.
We thank the reviewer for their valuable comments and suggestions. We agree that all the points raised by the reviewer are important in improving the quality of this manuscript and have been addressed to the best of our ability.
- Some minor English and grammatic mistakes should corrected throughout the manuscript.
We have corrected these issues to the best of our knowledge.
- In introduction, please add all types of SRB.
While other bacteria such as Fusobacterium nucleatum are capable of producing H2S through an assimilatory sulfate reduction pathway that results with cysteine as an end product, our review focuses only on the dissimilatory sulfate reduction pathway which is typical only to SRB, as carried out by DSV and 4 other genera. We have now included this in lines 56-60.
- What about the biotechnological uses of Desulfovibrio.
We have now added more information in lines 46-50 to include some uses of DSV. However, considering the scope of this review and keeping in mind the suggestions of the other reviewer to keep this manuscript focused on diseases, we have kept this section short and concise.
- There are any active constituents that are isolated from this type of bacteria. If yes, please add all and draw the chemical structures.
The main structural constituent reported to have been isolated from DSV is the lipopolysaccharide. The generic structure of LPS can be seen in figure 2. While reference 183 has reported LPS structure in D.desulfuricans, the authors have not provided detailed structures that can be represented in a figure format. We do not have the necessary knowledge to draw DSV LPS structure which is not available in the literature. We have now added reference 183 and added information related to DSV LPS structure in lines 482-485.
- Citation of references in the text is uncorrected example [13] [14] [15]...........please correct all as [13-15].
We have now corrected this.
- List of abbreviations with complete name should be provided.
We have now provided the complete names for all abbreviations the first time they appear in the text.
- Dear authors, any review should attract the readers by providing figures or equations for the mechanisms either for its benefits or hazards. So, please add some figures to the review to illustrate the bacterial mechanism in production of H2S, bacteremia, IBD, and others….., Just enumerating the benefits or hazards is not enough.
We have now provided figure 3 for the dissimilatory sulfate reduction pathway of H2S production. There are no known mechanisms of how DSV may cause bacteremia or IBD. But we have edited figure 2 to include the possible mechanisms by which DSV may contribute to diseases, possibly by its production of H2S , LPS and outer membrane vesicles (OMVs). To our knowledge, these are the only three mechanisms reported in the literature by which DSV has been suggested to possibly mediate it effects, with some evidence. We now also have added two tables to summarize the more direct effects of DSV invivo and invitro to make the review more cohesive.

Round 2
Reviewer 2 Report
No Comments.
Minor editing of English language required